# Raspberry Ketone-Mediated Inhibition of Biofilm Formation in *Salmonella enterica* Typhimurium—An Assessment of the Mechanisms of Action

**DOI:** 10.3390/antibiotics12020239

**Published:** 2023-01-23

**Authors:** Arakkaveettil Kabeer Farha, Zhongquan Sui, Harold Corke

**Affiliations:** 1Department of Biotechnology and Food Engineering, Guangdong Technion—Israel Institute of Technology, 241 Daxue Road, Shantou 515063, China; 2Department of Food Science & Technology, School of Agriculture and Biology, Shanghai Jiao Tong University, Shanghai 200240, China; 3Faculty of Biotechnology and Food Engineering, Technion–Israel Institute of Technology, Haifa 3200003, Israel

**Keywords:** *Salmonella* Typhimurium, Raspberry Ketone, biofilm, amyloid curli, quantitative proteomics

## Abstract

*Salmonella enterica* is an important foodborne pathogen that causes gastroenteritis and systemic infection in humans and livestock. *Salmonella* biofilms consist of two major components—amyloid curli and cellulose—which contribute to the prolonged persistence of *Salmonella* inside the host. Effective agents for inhibiting the formation of biofilms are urgently needed. We investigated the antibiofilm effect of Raspberry Ketone (RK) and its mechanism of action against *Salmonella* Typhimurium 14028 using the Congo red agar method, Calcofluor staining, crystal violet method, pellicle assay, and the TMT-labeled quantitative proteomic approach. RK suppressed the formation of different types of *Salmonella* biofilms, including pellicle formation, even at low concentrations (200 µg/mL). Furthermore, at higher concentrations (2 mg/mL), RK exhibited bacteriostatic effects. RK repressed cellulose deposition in *Salmonella* biofilm through an unknown mechanism. Swimming and swarming motility analyses demonstrated reduced motility in RK-treated *S.* typhimurium. Proteomics analysis revealed that pathways involved in amyloid curli production, bacterial invasion, flagellar motility, arginine biosynthesis, and carbohydrate metabolism, were targeted by RK to facilitate biofilm inhibition. Consistent with the proteomics data, the expressions of *csgB* and *csgD* genes were strongly down-regulated in RK-treated *S.* typhimurium. These findings clearly demonstrated the *Salmonella* biofilm inhibition capability of RK, justifying its further study for its efficacy assessment in clinical and industrial settings.

## 1. Introduction

*Salmonella enterica* is an important foodborne pathogenic bacterium which causes gastroenteritis and systemic infection in humans and in livestock. This bacterium represents a major public health challenge, and imposes a significant burden on the food industry [1]. One of the most notable features of *Salmonella* sp. Is their ability to form biofilms on biotic as well as abiotic surfaces, such as in the food-processing environment [2]. Biofilm formation allows *Salmonella* to overcome external harsh environmental pressures, and to prolong its colonization and survival. Furthermore, biofilm increases the tolerance of bacteria to antibiotics and disinfectants, due to the heterogeneous nature of bacterial cells within the biofilm [3,4]. Additionally, the establishment of biofilm promotes *Salmonella* transmission by an asymptomatic carrier through intermittent shedding into surroundings [5]. In this context, there is an urgent need for the development of novel strategies to prevent the multicellular behavior/biofilm formation of *Salmonella* in different environments.

Amyloid curli and cellulose are two major components of extracellular polymeric substances (EPS) matrix of *Salmonella* biofilms, which are attributed to the rdar (red dry and rough) morphotype of *Salmonella* on Congo red agar plate [6]. As a result of its high stability against proteolytic enzymes and detergents, the removal of curli-based biofilm is extremely difficult. The expression of two csg (agf) operons, viz. *csgBAC* and *csgDEFG*, is very important for curli production. Thus, interfering with curli biogenesis/assembly provides a unique opportunity to inhibit *Salmonella* biofilms [7,8,9]. Several plant-derived compounds, including cinnamaldehyde [10], furanones [11], carvacrol [12], thymol [13], and quercetin [14] have been shown to be effective against *Salmonella* biofilms. However, very few compounds have been demonstrated to target amyloid curli synthesis in *Salmonella* biofilm. Since *Salmonella* forms biofilms on the surfaces of host cells, using curli as the major constituent [15], it is critical to develop effective antibiofilm agents to hinder curli-dependent biofilm formation.

During a screening study for identifying novel natural compounds with antibiofilm potential against *Salmonella* sp., we identified Raspberry Ketone (RK; 4-(p-hydroxyphenyl)-2-butanone) as a promising candidate. RK, a flavoring agent commonly found in raspberries, has anti-obesogenic and melanogenesis-inhibiting properties [16,17]. Preclinical studies indicated that RK supplementation with a calorie-restricted diet can abrogate obesity-induced Alzheimer’s disease [18]. Since our initial screening studies indicated the propensity of RK to inhibit biofilm formation, we decided to conduct a more detailed study on RK. To best of our knowledge, there are no previous studies concerning the activity of RK on *Salmonella* biofilm. Therefore, the present study explored the effect of RK on biofilm formation by *Salmonella*. Additionally, proteomics analysis and gene expression level assessments were performed to obtain deeper insight into the RK-specific mechanisms of *Salmonella* biofilm inhibition.

## 2. Results

### 2.1. Anti-Bacterial Effect of RK

The anti-bacterial effect of RK was assessed using a microbroth dilution method (Figure 1A). The minimum inhibitory concentration (MIC value) of RK against *S.* typhimurium 14028 was 2 mg/mL. To determine whether the antibiofilm effect of RK was not due to its growth inhibitory effect, the growth of *S.* typhimurium was analyzed in the presence of sub-MIC doses of RK. As shown in Figure 1B, no substantial differences were observed in the cell densities between RK-treated (sub-MIC, below 500 μg/mL) and control *S.* typhimurium, suggesting that RK did not show any anti-bacterial activity against *S.* typhimurium at lower concentrations.

### 2.2. RK Inhibits Biofilm Formation of S. Typhimurium

Generally, *Salmonella* forms different biofilm types in different conditions [19]. *Salmonella* can form biofilm on LB without salt agar, with characteristic surface patterns of the red, dry, and rough (rdar) morphotype. The effect of RK on the rdar morphotype was studied using the Congo red agar plate method. In the control group, after 4 days of incubation at 28 °C, *S.* typhimurium formed colonies exhibiting rdar morphology, whereas in the presence of RK (200 μg/mL), *S.* typhimurium showed loss of its rdar morphology, and the colony appeared to be smooth and less reddish in color (Figure 2A). The effect of RK on the rdar morphotype of other *S. enterica* strains (*S.* Enteritidis ATCC 13076 and *S.* Enteritidis SJTUF 10987) was also assessed. The results revealed that both *Salmonella* strains formed colonies without any wrinkled pattern on the surface on Congo red agar plate, indicating that the effect of RK on the rdar morphology of *Salmonella* was not strain-specific (Appendix A).

*Salmonella* develops biofilm in liquid cultures, known as a pellicle, at the air–liquid interface [20]. A dense pellicle was observed in control wells, whereas a significantly thinner layer of pellicle was observed in RK-treated *S.* typhimurium (Figure 2B). In liquid culture under microaerophilic conditions, *S.* typhimurium forms multicellular aggregates and planktonic cells. After 48 h treatment with RK, the number and thickness of multicellular aggregates was reduced (Figure 2C). Light microscopic analysis of the morphology of *S.* typhimurium cell aggregates grown in control flask showed tightly attached and highly compact large cell aggregates. In the presence of RK, the planktonic cells were visible with fewer small cell aggregates, suggesting the biofilm inhibitory potential of RK (Figure 2D). *S.* typhimurium produced robust biofilm in submerged conditions in a 96-well polystyrene plate, as evidenced by crystal violet staining. Treatment with RK resulted in a reduction in biofilm biomass, in a dose-dependent manner (Figure 2F). Light microscopic observation of biofilm formation after treatment with RK showed significantly lower levels of biofilm formation, indicating its antibiofilm potential against *Salmonella* sp. (Figure 2E). These results indicate that RK has potential to inhibit the development of biofilm in different settings by *Salmonella*.

### 2.3. RK Reduces Cellulose Deposition in S. Typhimurium Biofilm

The cellulose content of *S.* typhimurium biofilms was analyzed using the Calcofluor staining method. In the control group, *S.* typhimurium exhibited good amounts of cellulose during biofilm matrix formation, as evidenced from the intense fluorescence exhibited by the untreated colonies under UV. After treatment with 200 μg/mL RK, the fluorescent intensity was significantly reduced, indicating a reduction in cellulose in the EPS matrix of *S.* typhimurium biofilm (Figure 3A,B). Though not statistically significant, treatment with 200 μg/mL concentration of RK itself was adequate to reduce *Salmonella*-mediated biofilm cellulose production by 50%.

### 2.4. RK Affects Motility of S. Typhimurium

*S.* typhimurium exhibited higher motility in the control group, as compared to the RK-treated group. At a concentration of 200 μg/mL, RK reduced the swimming and swarming motility of *S.* typhimurium (Figure 3C).

### 2.5. RK Treatment Inhibits the Expression of Proteins Essential for S. Typhimurium Biofilm Development

To understand the mechanism behind the RK-mediated inhibition of biofilm formation, we conducted proteomic studies. To quantify the differentially expressed proteins (DEP) of *S.* typhimurium upon RK treatment for 24 h, a TMT-based quantitative proteomic method using mass spectrometry (LC-MS) was performed. A total of 3003 proteins were identified, and DEP analysis revealed 60 and 68 up-regulated and down-regulated proteins, respectively (Figure 4A,B and Appendix A). GO annotation analysis categorized DEP to identify their localization in cellular components (CC), molecular functions (MF) and biological processes (BP) (*p*-value < 0.05) (Figure 4C). GO enrichment analysis showed that DEP were enriched in an L-threonine catabolic process to propionate in terms of biological processes (*p* = 0.001). GO analysis of molecular functions indicated that DEP were enriched in ornithine carbamoyltransferase activity (*p* = 0.0003). GO annotation of cellular component indicated that DEP were enriched in bacterial type flagellum-hook (*p* = 0.0001). GO-enrichment analysis of up-regulated and down-regulated DEP is provided in Appendix A.

KEGG pathway analysis showed that *Salmonella* infection, propanoate metabolism (carbohydrate metabolism), flagellar assembly, bacterial invasion of epithelial cells, and arginine biosynthesis pathways were enriched after exposure to RK (Figure 4D and Appendix A). Proteins that are involved in flagellar assembly, including FlgK, FlgL, FliD, FliL, FliZ, and MotA, and proteins in arginine metabolism (ArcA, ArgF, and ArcC) were significantly down-regulated upon RK treatment. Among down-regulated proteins, SipA, SipB, SipC, SopB, SopE2, and SptP, six type III effectors of *Salmonella* pathogenicity island 1 (SPI-1), were also included. The expressions of proteins involved in propionate metabolism, including formate C-acetyltransferase (*tdcE*), 2-methylcitrate dehydratase (*prpD*), citrate synthase, 2-methylisocitrate lyase (*prpB*) and propionate kinase (*tdcD*), were reduced in *S.* typhimurium in response to RK treatment.

The proteins associated with amyloid curli synthesis, such as CsgB, CsgD, CsgC, CsgG, and CsgF, were significantly down-regulated in RK-treated *S.* typhimurium biofilm compared to the control. In particular, the minor curlin subunit CsgB exhibited a 0.12-fold change reduction after RK treatment (Appendix A). Additionally, the expressions of several proteins were found to be up-regulated after RK treatment. The expressions of proteins, including cytochrome c-type protein (*tor C*), chaperone protein TorD (*torD*) and trimethylamine-N-oxide reductase (*torA*) in *torCAD* operon, were up-regulated (3.1-, 3.2-, and 3.8-fold changes) (Appendix A).

A protein–protein interaction (PPI) network was constructed, and hub molecules were identified based on the degree rank to understand the possible molecular mechanisms of RK on *S.* typhimurium biofilm development. Results showed that the top ten hub molecules were Ig-like domain repeat protein (G2712_12205, degree = 24), secreted effector protein SptP (*sptP*, degree = 23), cell invasion protein SipB (*sipB*, degree = 21), SIS domain-containing protein (G2712_03435, degree = 21), UPF0319 protein YccT (*yccT*, degree = 19), protein-secreting ATPase (CBS77_02345, degree = 19), minor curlin subunit (*csgB*, degree = 19), chaperone protein TorD (*torD*, degree = 19), TolC family protein (GJG76_22260, degree = 18), and NAD-dependent dihydropyrimidine dehydrogenase subunit PreA (*preA*, degree = 18) (Figure 4F). The expressions of G2712_12205, SptP, SipB, YccT, protein-secreting ATPase, CsgB, and TolC family protein were down-regulated, whereas the protein expressions of SIS domain-containing protein, TorD, and PreA were up-regulated by 1.8-, 3.1-, and 2.1-fold, respectively, in RK-treated *S.* typhimurium (Appendix A).

### 2.6. RK Inhibits the Expression of Biofilm-Related Genes in S. Typhimurium

The transcript level expressions of selected gene targets in *S.* typhimurium biofilm matrix—*csgD* and *csgB*—were assessed to validate the proteomics data. In line with the findings from the proteomics analysis, the transcript levels of *csgD* and *csgB* were found to be down-regulated in *S.* typhimurium by RK treatment (Figure 4E).

## 3. Discussion

In this study, we investigated the effect of RK on *S.* typhimurium biofilm. RK exhibited a bacteriostatic effect at higher concentrations (≥2 mg/mL), whereas treatment with 200 μg/mL RK prevented the formation of *S.* typhimurium biofilm. As a dietary supplement and a GRAS (Generally Recognized as Safe) ingredient, the daily recommended dose of RK is between 100 and 1400 mg [21]. Consequently, RK can be utilized as a preventive supplement to prevent *Salmonella* biofilm formation. RK’s relative safety, easy accessibility, and capacity for sustainable mass production using chemical and biotechnological synthesis methods are its main advantages as an antibiofilm agent.

In order to assess whether RK has the ability to inhibit different biofilm types of *S.* typhimurium in liquid and solid media, *S.* typhimurium was grown in different settings, such as in standing liquid culture, microaerophilic flask culture [22], as a monoculture biofilm in a 96-well polystyrene plate, and on Congo red agar plates [6]. *S.* typhimurium developed a robust biofilm in all of these growth conditions, which was inhibited by RK treatment. These data suggest that RK prevents *Salmonella* biofilm, irrespective of the growth settings, without affecting planktonic growth. Furthermore, regardless of strain specificity, RK inhibited biofilm development in *Salmonella*. Cellulose is an integral part of pellicle formation [23]. Upon incubation of *S.* typhimurium with RK, a reduction in Calcofluor fluorescence was observed, which correlated with inhibition of cellulose production in the biofilm. However, cellulose synthesis regulatory protein diguanylate cyclase (*yedQ*) and genes in the cellulose biosynthesis operons *bcsABZC* and *bcsEFG* did not show any significant difference in their expressions after treatment with RK (*p* > 0.05). In a similar study, reduced cellulose in the biofilm matrix of *Escherichia coli* K-12 strains was observed when treated with quercetin, without inhibiting the expression of *dgcC*, a gene that encodes a diguanylate cyclase required for cellulose biosynthesis [6]. Thus, it is possible that RK interferes with the deposition of cellulose in the biofilm without interfering with its production. However, additional studies are required to understand the exact mechanism.

GO analysis and KEGG pathway analysis revealed that DEP were enriched in the pathways of *Salmonella* infection, propanoate metabolism, bacterial invasion of epithelial cells, arginine biosynthesis, and flagellar assembly. The results indicated that RK can interfere with multiple processes in *S.* typhimurium. The proteins in the arc operon are highly expressed in the biofilms of many bacteria to regulate intracellular pH, and mutations in the arc operon reduce biofilm formation [24]. Bacterial cells within the biofilm matrix are exposed to hypoxic conditions. Under anaerobiosis, *Salmonella* uses the arginine metabolic pathway to produce energy [25]. Our results showed that RK could suppress the expression of proteins involved in arginine biosynthesis, including the arginine deiminase (*arcA*) protein. A similar effect has been observed in multidrug-resistant *Pseudomonas aeruginosa* biofilms treated with silver nanoparticles [26].

Curli are the proteinaceous thin filaments produced by *Salmonella* sp. involved in cell-to-cell interaction and surface colonization [7]. The upstream regulator CsgD activates the production of curli by transcriptional activation of the *csgBAC* operon that encodes the structural genes of curli [27], while CsgB, the nucleator protein, is involved in the folding and assembly of curli amyloid fiber, CsgA. The export of Csg proteins to the cell surface is guided by the accessory proteins CsgG, CsgE, and CsgF, whereas CsgC acts as an inhibitor that prevents intracellular amyloid formation [7]. We found that as a result of the antibiofilm effects of RK, the expression levels of amyloid proteins, except CsgA in *csgBAC* and CsgE in the *csgDEFG* operon, were significantly down-regulated. Previous studies showed that bacterial curli amyloid fibrils resemble the Aβ fibrils found in Alzheimer’s disease [8]. RK has been shown to decrease accumulation of Aβ plaques in obesity-induced Alzheimer’s disease rats, and to inhibit amyloidogenesis [18]. In our study, we found that RK was extremely effective at inhibiting bacterial amyloid production.

Flagellar motility plays a critical role in the development of *Salmonella* biofilms [28,29]. Flagellar motor stator protein MotA is necessary for the rotation of flagella, and impairment of this gene completely inhibits bacterial motility. Remarkably, the expression of motility-associated proteins such as hook-filament junction proteins (FlgK and FlgL), the filament (FliC), and the filament cap (FliD) in *S.* typhimurium, were down-regulated by RK treatment. Moreover, FliL, a protein required for swarming motility [30], was also suppressed. These results explain the mechanism by which RK reduced the swimming and swarming motility in *Salmonella*, thereby inhibiting the development of *Salmonella* biofilms.

*S.* typhimurium utilizes a type III secretion system to translocate effector proteins into host cells. The transport of bacterial effector proteins across the host cell membrane is facilitated by forming a translocation complex consisting of SipB, SipC, and SipD effector proteins. SopB, SopE2, and SipA are the SPI-1-secreted effectors responsible for the disruption of tight junction structure and function [31]. These secreted effectors modulate the functions of host cells, and activate specific signalling cascades that result in the production of pro-inflammatory cytokines and intestinal inflammation. Notably, many effector proteins were down-regulated, including SptP, when exposed to RK.

The operon *torCAD*, encoding the trimethylamine N-oxide (TMAO) respiration system, enables the use of TMAO as a terminal electron acceptor for respiration when oxygen is not available [32]. Zafar et al. [22] found that when exposed to clarithromycin antibiotic, the expressions of three genes (*tor C*, *torA*, and *tor D*) in the *torCAD* operon of *S.* typhimurium UMR1 were up-regulated. Consistent with this study, the expressions of TorC, TorD, and TorA proteins were up-regulated in RK-treated *S.* typhimurium. The overexpression of *torCAD* operon proteins reflected the *S.* typhimurium response to the oxygen depletion caused by RK treatment. The genes involved in propanoate metabolism were found to be up-regulated in *Salmonella* biofilms, indicating their importance in biofilm development [33,34]. Compared with the control biofilm, the expressions of 2-methylcitrate dehydratase and 2-methylisocitrate lyase proteins were suppressed in RK-treated *S.* typhimurium. These results showed that RK could affect propanoate metabolism to exert its antibiofilm action in *S.* typhimurium.

## 4. Materials and Methods

### 4.1. Bacterial Strains and Growth Conditions

*Salmonella enterica* Typhimurium ATCC 14028, *Salmonella enterica* Enteritidis ATCC 13076, and *S. enterica* Enteritidis SJTUF 10987 were used for the study. Bacteria were grown in tryptic soy broth (TSB). For biofilm studies, overnight bacterial cultures of *Salmonella* were diluted in Luria Bertani (LB) without salt medium.

### 4.2. Determination of MIC

The MIC of RK against *S.* typhimurium was determined with a broth dilution method based on the Clinical and Laboratory Standards Institute guidelines (CLSI 2009) [35]. RK was double diluted in Mueller Hinton (MH) broth to obtain a concentration range of 0–4000 μg/mL in a 96-well plate. To each well, 100 μL of bacterial culture, with a final bacterial concentration of approximately 10^6^ CFU/mL, was added. MH broth containing 0.1% DMSO was used as the negative control. After 24 h of incubation at 37 °C, the lowest concentration of RK that resulted in no visible bacterial growth was considered as MIC.

### 4.3. Growth Curves for Salmonella

S. Typhimurium was grown in LB media with RK (0–1000 μg/mL) and incubated at 37 °C with shaking at 150 rpm. The optical density was measured with 1 h interval at 600 nm using microtiter plate reader (Molecular Devices, San Jose, CA, USA).

### 4.4. Congo Red Agar Plate Assay

The overnight culture of *S.* typhimurium (2 µL) was spotted on LB agar without salt, and supplemented with Congo red (40 µg/mL) and Coomassie Brilliant Blue (20 µg/mL). Colony morphology was analyzed after 96 h of incubation at 28 °C [6].

### 4.5. Calcofluor Staining Assay

The overnight culture of *S.* typhimurium (5 µL) was spotted on LB agar without salt, and supplemented with Calcofluor stain (200 μg/mL). Plates were incubated at 28 °C for 48 h before being examined for fluorescence using a UV transilluminator. The cellulose production was quantified as previously described, with minor modifications [36]. Briefly, the overnight culture of *S.* typhimurium (8 µL) was spotted on LB agar without salt, and supplemented with 50 μg/mL Calcofluor stain in a black 96-well microtiter plate with a clear bottom (BD Falcon). The emission intensity at 460 nm (excitation at 355 nm) was measured after 48 h of incubation at 28 °C.

### 4.6. Pellicle Formation Assay

To evaluate the effect of RK on pellicle formation by *S.* typhimurium, bacteria were grown in a 24-well plate containing 1.8 mL of LB without salt media, with 200 μL *S.* typhimurium and RK (200 μg/mL), and incubated for 96 h at 28 °C. After 4 days of incubation, the formation of pellicles was observed visually, and assessed using a pipette tip.

### 4.7. In Vitro Flask Model Biofilm Development

*S.* typhimurium was incubated with RK (200 μg/mL) in LB without salt media for 24 h, with agitation at 150 rpm at 28 °C. At the end of incubation, cell aggregates were identified, and the morphology of the aggregates was observed with light microscopy (10× magnification) [22].

### 4.8. Crystal Violet (CV) Staining Assay

To investigate the effect of RK on *S.* typhimurium biofilms, bacteria were grown in LB without salt medium, containing sub-MIC concentrations of RK for 48 h at 28 °C in a 96-well microtiter plate. A crystal violet staining assay was performed to quantify the biofilm, as previously described [37]. Briefly, after treatment with RK, the supernatant was removed, and wells were washed with phosphate-buffered saline (PBS). After that, the adherent cells were stained with 0.1% CV (200 μL) and incubated for 30 min at room temperature. The excess stain was removed, and CV was dissolved in DMSO (200 μL). The absorbance was measured at 595 nm.

### 4.9. Light Microscopy

*S.* typhimurium biofilms attached to the bottom of the 96-well microtiter plate were washed three times with PBS, stained with 0.4% CV solution, and analyzed with light microscopy (10× magnification).

### 4.10. Motility Assay

The overnight culture of *S.* typhimurium was spotted on swim (0.3% agar) plates or on swarm (0.5% agar) plates, without RK (control) or containing RK (200 μg/mL). All plates were incubated at 37 °C for 24 h [38].

### 4.11. TMT-Labeled Quantitative Proteomic Analysis

#### 4.11.1. Protein Extraction and Trypsin Digestion

*S.* typhimurium was incubated in LB without salt media, with or without RK, with agitation at 150 rpm. After 24 h of incubation at 28 °C, the biofilms were obtained by centrifugation at 5000× *g* for 10 min. Biofilms were rinsed with PBS, and resuspended in PBS. The cell pellets were mixed with SDS lysis buffer (Beyotime Biotechnology, Shanghai, China) and PMSF (phenylmethanesulfonylfluoride), and ultrasonicated for 3 min (power 80 W). The samples were centrifuged at 12,000× *g* for 10 min at room temperature to collect the supernatant. The protein concentration was determined using a BCA Protein Assay Kit (ThermoFisher, Waltham, MA, USA). The proteins were separated using 12% SDS-PAGE electrophoresis gel, and proteins were stained with Coomassie Brilliant Blue. For trypsin digestion, the total protein (50 μg) was incubated with 5 mM DTT at 55 °C for 30 min. After 15 min of incubation at room temperature with 10 mM iodacetamide (IAA), 6 times the volume of acetone was added and incubated further overnight at −20 °C. The samples were centrifuged at 8000× *g* for 10 min at 4 °C to collect the precipitated proteins. Subsequently, proteins were resuspended in 200 mM TAEB (triethylammonium bicarbonate buffer; Sigma-Aldrich, Shanghai, China) and incubated overnight at 37 °C after the addition of trypsin (1 mg/mL) (protein (g):trypsin(g), 50:1). The samples were lyophilized after enzymatic hydrolysis.

#### 4.11.2. TMT Labelling and Reversed-Phase Chromatographic Separation

The digested peptides were suspended in 100 mM TEAB, and were labelled with TMT reagent (ThermoFisher, Fairlawn, NJ, USA) at room temperature for 60 min. The peptides were subjected to fractionation via a reversed-phase high-performance liquid chromatography (Agilent 1100 HPLC, Agilent, Palo Alto, CA, USA) system using an Agilent ZORBAX 300Extend-C18 column (2.1 I.D × 150 mm length, 5 μm particle size). Solvent A consisted of acetonitrile (ACN) with water (2:98, *v*/*v*), and solvent B was composed of ACN with water (90:10, *v*/*v*); the flow rate was maintained at 300 μL/min. The gradient elution conditions were 0~8 min, 98% A; 8~8.01 min, 98~95% A; 8.01~48 min, 95~75% A; 48~60 min, 75~60% A; 60~60.01 min, 60~10% A; 60.01~70 min, 10% A; 70~70.01 min, 10~98% A; 70.01~75 min, 98% A. The fractions were collected and lyophilized.

#### 4.11.3. LC-MS Analysis

The peptides were analyzed using an EASY-nLC 1000 ultra-high pressure liquid chromatography system coupled to a Q Exactive Mass Spectrometer (ThermoFisher, Fairlawn, NJ, USA). Briefly, the samples were loaded at a flow rate of 3 μL/min to the pre-column Acclaim PepMap100 (100 μm × 2 cm, RP-C18, ThermoFisher), and separated by the Acclaim PepMap100 (75 μm × 25 cm, RP-C18, ThermoFisher). Solvent A consisted of water (98%)-CAN (1.9%)-formic acid (0.1%), and solvent B consisted of water-ACN-formic acid (90:9.9:0.1; *v*/*v*/*v*). The gradient elution conditions were as follows: 0~38 min, 6–22% B; 38~50 min, 22–36% B; 50~54 min, 36–85% B; 54~60 min, 85% B. The mass spectrometry settings were set as follows: the first-level MS mass resolution was 70,000, automatic gain control value was 1 × 10^6^, and the maximum injection time was 50 ms. The mass spectrometry scan was a full scan in the range of 300–1600 *m*/*z*, and MS/MS scanning was performed for 10 of the highest peaks. The resolution of MS/MS was 17,500, the automatic gain control was 2 × 10^5^, and the maximum ion accumulation time was 80 ms. MS/MS acquisitions were completed using high-energy collision cleavage in data-dependent positive ion mode, and the collision energy was 32. The dynamic exclusion time was 30 s.

#### 4.11.4. Data Acquisition and Bioinformatics Analysis

The raw data were analyzed using Proteome Discover 2.4 (ThermoFisher). The specific search parameters were set as follows: static modification, TMT (N-term, K); carbamidomethyl(C); dynamic modification, oxidation(M), acetyl(N-term); MS1 tolerance, 10 ppm; MS2 tolerance, 0.02 Da; missed cleavages, 2; digestion, trypsin; database, uniprot-taxonomy_149539.fasta; false-discovery rate (FDR), <0.01; fold change = above 1.5 or below 0.83; and *p*-value < 0.05. For the identified proteins, the annotated information was extracted based on Uniprot, KEGG, GO, and KOG/COG and other databases, in order to mine the protein function. Additionally, the protein–protein interaction network diagram could also be presented through Cytoscape software.

### 4.12. RNA Extraction and Reverse Transcription–Quantitative PCR (RT–qPCR)

To determine the effect of RK on the transcription of curli-related genes, *S.* typhimurium was grown with or without RK at 28 °C for 24 h. Total RNA was extracted with the Tiangen RNAprep Pure Cell/Bacteria Kit (Tiangen, Beijing, China), according to the manufacturer’s instructions. The RNA quality and concentration were determined with a nucleic acid and protein spectrophotometer (Nano-200, Aosheng Instrument, Hangzhou, China). The Takara PrimeScript™ RT Reagent Kit (Takara, Kyoto, Japan) was used to reverse transcribe the RNA into cDNA, according to the manufacturer’s instructions. The cDNA samples were stored at −20 °C until analysis. The primer sequences that were used for RT–qPCR are listed in Appendix A. The qPCR reactions (25 μL) with SYBR^®^ Premix Ex Taq™ II (Takara, Kyoto, Japan) were performed using the Roche LightCycler^®^ 96 with the following conditions: initial denaturation at 95 °C for 30 s, followed by 40 cycles of denaturation at 95 °C for 5 s, annealing at 60 °C for 30 s, and a dissociation step at 95 °C for 15 s and 60 °C for 30 s. The relative gene transcription in the samples was analyzed using the 2^−ΔΔCt^ method, relative to the reference gene 16s rRNA.

### 4.13. Statistical Analysis

All experiments were carried out in three biological replicates. The data are presented as mean ± standard deviations (SD). Statistical analyses were performed using GraphPad Prism 9 (Graph Pad Software Inc., San Diego, CA, USA). Statistical significance for antibiofilm assays (crystal violet staining assay and Calcofluor staining assay) was assessed using one-way analysis of variance (ANOVA), with Dunnett’s multiple comparison post hoc test comparing biofilm formation in the presence of RK to biofilm formation in the absence of RK. To determine the significance between the control and treated sample in the gene expression study, paired Student’s *t*-test was used. A *p*-value < 0.05 was considered statistically significant. * *p*  <  0.05, ** *p*  <  0.01, *** *p*  <  0.005, **** *p*  <  0.001.5.

## 5. Conclusions

Our results show that RK inhibits *Salmonella* biofilm development. Proteomic analysis suggests that RK interferes with multiple cellular processes to combat biofilm formation, including curli synthesis, flagellar motility, type III SPI-1 effector proteins, arginine biosynthesis, and carbohydrate metabolism. RK affects amyloid curli synthesis by inhibiting the expressions of curli-related proteins such as CsgB and CsgD. These findings provide insights into the mode of action of RK, which may be further developed as an approach to prevent biofilm formation by *Salmonella* sp. To assess RK’s efficacy in preventing *Salmonella* biofilms on food contact surfaces and in food model systems, additional research is required. Additionally, more research in animal models should be conducted to confirm the clinical efficacy of RK. If the findings of the current study are properly extended, they may have far-reaching applications in the food and healthcare sectors.

## Figures and Tables

**Figure 1 antibiotics-12-00239-f001:**
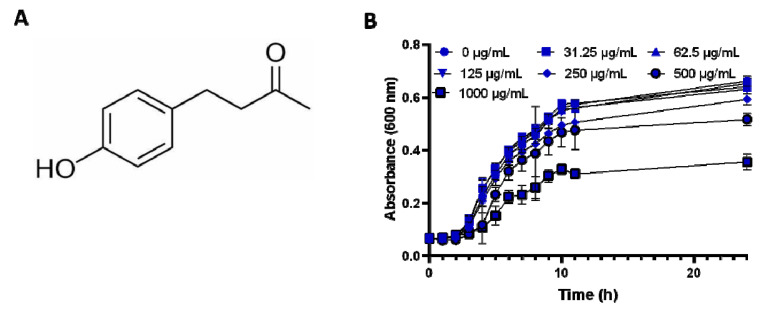
Antibacterial effect of Raspberry Ketone (RK): (**A**) structure of RK; (**B**) growth curve analysis of *S.* typhimurium in the presence of varying concentrations of RK. Data are presented as mean ± SD.

**Figure 2 antibiotics-12-00239-f002:**
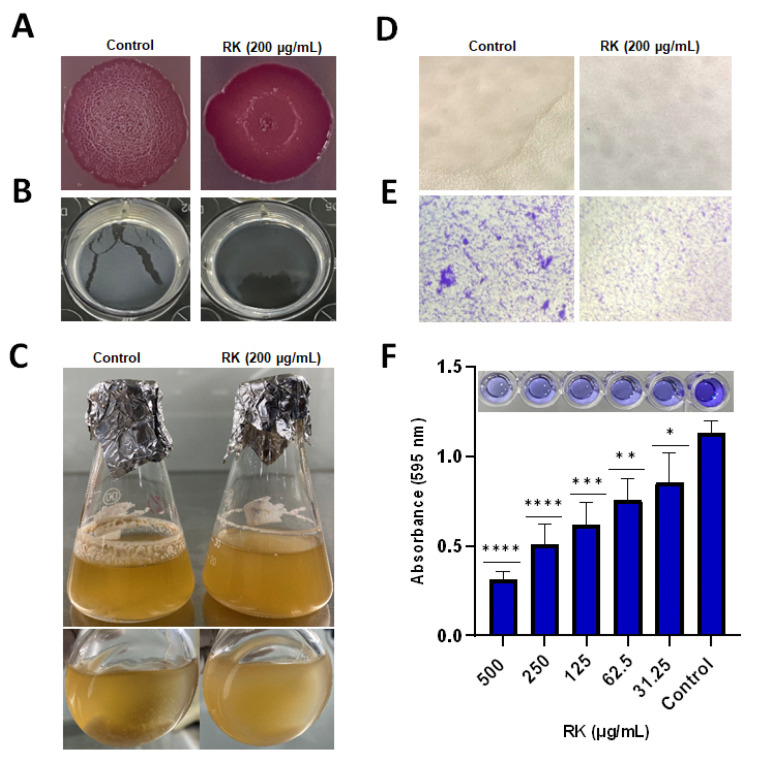
RK inhibits biofilm formation of *S.* typhimurium: (**A**) effects of RK on rdar morphotype of *S.* typhimurium; (**B**) pellicle formation inhibitory effects of RK; (**C**) effect of RK on cell aggregates formed by *S.* typhimurium under microaerophilic conditions; (**D**) light microscopy images of cell aggregates; (**E**) light microscopy images of control and RK-treated *S.* typhimurium; (**F**) antibiofilm effect of RK against *S.* typhimurium in a 96-well plate by CV staining assay. Data are presented as mean ± SD, * *p*  <  0.05, ** *p*  <  0.01, *** *p*  <  0.005, **** *p*  <  0.001, compared to control group.

**Figure 3 antibiotics-12-00239-f003:**
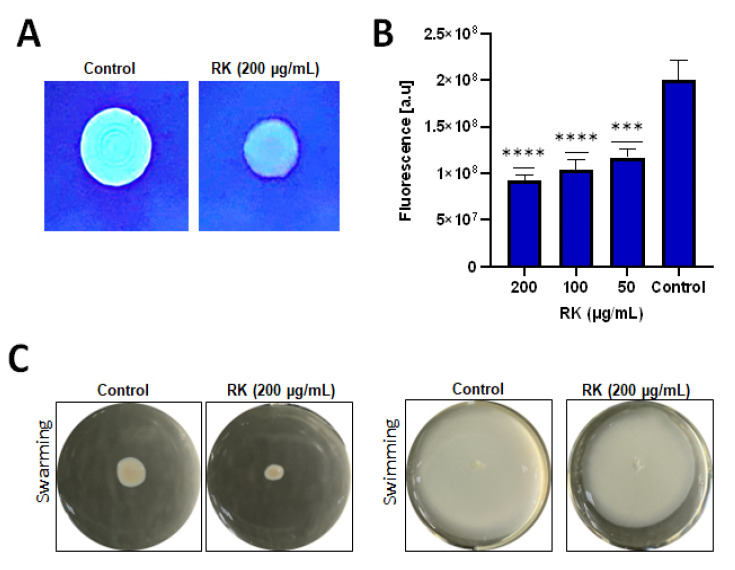
Inhibition of biofilm cellulose production and motility of *S.* typhimurium by RK: (**A**) representative images of *S.* typhimurium stained with Calcofluor staining; (**B**) evaluation of biofilm cellulose production inhibition in RK-treated cells by measuring fluorescence; (**C**) swimming and swarming motility inhibition of *S.* typhimurium at 200 μg/mL concentration of RK. Data are presented as mean ± SD, *** *p*  <  0.005, **** *p*  <  0.001, compared to control group.

**Figure 4 antibiotics-12-00239-f004:**
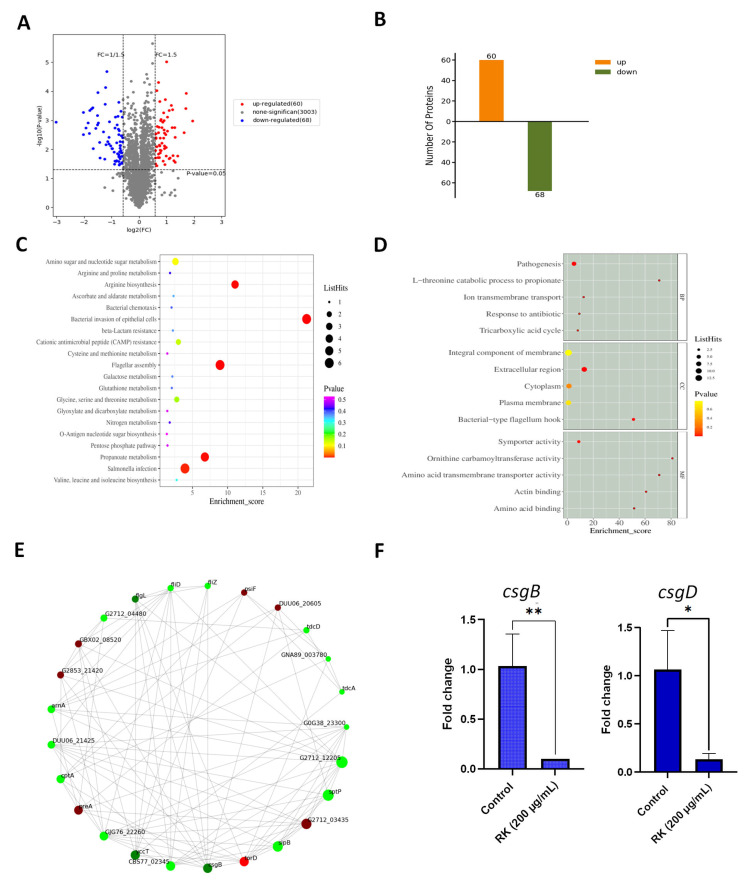
Proteomic analysis of *S.* typhimurium upon RK treatment: (**A**) volcano plot of the differentially expressed proteins (DEP); (**B**) numbers of DEP; (**C**) GO enrichment analysis of DEP; (**D**) KEGG pathway-based enrichment analysis of DEP; (**E**) protein–protein interaction (PPI) network of DEP; (**F**) effect of RK on the expression of *S.* typhimurium biofilm-related genes. Data are presented as mean ± SD, * *p*  <  0.05, ** *p*  <  0.01, compared to control.

## Data Availability

Not applicable.

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
