# Peer review of "Raspberry Ketone-Mediated Inhibition of Biofilm Formation in Salmonella enterica Typhimurium—An Assessment of the Mechanisms of Action"

_antibiotics, 2023, doi:10.3390/antibiotics12020239_

Round 1
Reviewer 1 Report
Dear Author, I reviewed the manuscript (antibiotics-2163410) entitled Raspberry Ketone-Mediated Inhibition of Biofilm Formation in Salmonella enterica Typhimurium – An Assessment of the Mechanisms of Action. This manuscript presents relevant information about the antibiofilm potential of raspberry ketones against Salmonella Typhimurium. However, some sections of the presented data can be improved. For this reason, I consider that this manuscript needs minor changes to be considered for publication in this journal.
Additional comments.
Highlight the advantages of raspberry ketones as antibiofilm treatments.
Check the paragraph extension in this manuscript.
Try to include details in microbiological assays performed in this manuscript.
Include an experimental design that contains statistical factors, replicates, and variables of response in the statistical analyses applied to the findings of this research.
Try to discuss a possible ketones mode of action against Salmonella virulence or quorum sensing.
Include future trends to keep working with the obtained data.
Try to conclude with a general statement of the most relevant part of this study.
Author Response
Reviewer 1:
Dear Author, I reviewed the manuscript (antibiotics-2163410) entitled Raspberry Ketone-Mediated Inhibition of Biofilm Formation in Salmonella enterica Typhimurium – An Assessment of the Mechanisms of Action. This manuscript presents relevant information about the antibiofilm potential of raspberry ketones against Salmonella Typhimurium. However, some sections of the presented data can be improved. For this reason, I consider that this manuscript needs minor changes to be considered for publication in this journal.
We would like to thank the reviewer for a careful and thorough reading of this manuscript and for the thoughtful comments and constructive suggestions, which help to improve the quality of our manuscript. Following the reviewer’s comments, we have corrected and improved the manuscript. We hope that you find our responses satisfactory and that the manuscript is now acceptable for publication.
Here is a point-by-point response to the reviewer's comments and concerns.
Additional comments.
Question 1: Highlight the advantages of raspberry ketones as antibiofilm treatments.
Reply: Thank you for this suggestion. Following the reviewer’s comment, we have added a section explaining the advantages of Raspberry Ketone as an antibiofilm agent. Line 215-217.
Question 2: Check the paragraph extension in this manuscript.
Reply: Thank you for this suggestion. We have made changes in the paragraph extension.
Question 3: Try to include details in microbiological assays performed in this manuscript.
Reply: We appreciate the reviewer’s comment. We have made changes in the text. Line 335-339.
Question 4: Include an experimental design that contains statistical factors, replicates, and variables of response in the statistical analyses applied to the findings of this research.
Reply: Thank you for this suggestion. Following the reviewer’s comment, we have added information regarding the statistical analysis of the data. Line 425-433.
Question 5: Try to discuss a possible ketones mode of action against Salmonella virulence or quorum sensing.
Reply: Thank you for your suggestion.
Quorum sensing (QS), cell-cell communication mechanism, plays a key role in the development of biofilm in Salmonella sp. (Rana et al., 2021). S. Typhimurium possesses virulence-enhancing QS genes, such as SdiA, LuxS, and LsrA. (Alni et al., 2020). Further, genes like sdiA and adrA are involved in the production of extracellular polymeric substances (EPS) in Salmonella biofilm, which is known to be regulated by quorum sensing (Rana et al., 2021). In the current study, RK treatment had no discernible effect on the protein expression levels of luxS (FC=1.01), adrA (yedQ) (FC=1.03), and lsrA (FC=0.89) when compared to the control. However, these identified proteins were not differentially expressed. These findings demonstrated that in S. Typhimurium, RK has no or little impact on genes related to QS. In the current study, our main focus is on RK's inhibitory effect on the production of EPS, specifically curli and cellulose. We excluded the effect of RK on QS in Salmonella from this study for the reasons mentioned above.
Question 6: Include future trends to keep working with the obtained data.
Reply: We have made changes in text. Line 445-447
Question 7: Try to conclude with a general statement of the most relevant part of this study.
Reply: We have made changes in text. Line 441-442
References:
Rana, K., Nayak, S. R., Bihary, A., Sahoo, A. K., Mohanty, K. C., Palo, S. K., Sahoo, D., Pati, S., & Dash, P. (2021). Association of quorum sensing and biofilm formation with Salmonella virulence: story beyond gathering and cross-talk. Archives of microbiology, 203(10), 5887–5897. https://doi.org/10.1007/s00203-021-02594-y
Arch Microbiol. 2021;203(10):5887-5897.
Alni, R. H, Ghorban, K., Dadmanesh, M. (2020). Combined effects of Allium sativum and Cuminum cyminum essential oils on planktonic and biofilm forms of Salmonella typhimurium isolates. 3 Biotech, 10(7), 315. https://doi.org/10.1007/s13205-020-02286-2 3 Biotech. 2020;10(7):315.
Reviewer 2 Report
Authors investigated the anti-biofilm effects of RK on S. Typhimurium biofilm. This effect was studied both overall and in a specific manner to uncover the underlying mechanism. Authors obtained a lot of data for antibiofilm activity in various conditions, but also have shown which molecular components of biofilm formation process could be affected. The manuscript is generally well-written. Results are sufficiently described. I am only confused in the case of Figure 2 and its description in the text. Fig 2C in the figure is CV staining and its absorbance, but in the text it is Fig2C is a reference to light microscopy and cell aggregates - so please correct that.
Authors applied a lot of different techniques, which are usually suficciently described. I have few issues though:
- in MIC determinations - did you check this "visible" effect by eye or in some reader? In 96-well plate testing MIC by eye would be tricky in my opinion - I would reccomend using plate reader to measure OD with blank control adjusted to starting cell suspention. Also have the bacterial suspentions were done in MH medium?
- what concentrations of RK were used for growth curves?
- For protein extraction what medium was used to culture bacteria? LB?
Please, check the english for spelling and grammar: e.g., line 14 (abstract) not consist but contains (or consist of); line 293 (methods) bacterial solution was added
The obtained results are fully discussed and the conclusions are supported by the results. It leaves an interesting scientific niche for further studies.
Author Response
Reviewer 2:
Authors investigated the anti-biofilm effects of RK on S. Typhimurium biofilm. This effect was studied both overall and in a specific manner to uncover the underlying mechanism. Authors obtained a lot of data for antibiofilm activity in various conditions, but also have shown which molecular components of biofilm formation process could be affected. The manuscript is generally well-written. Results are sufficiently described. I am only confused in the case of Figure 2 and its description in the text. Fig 2C in the figure is CV staining and its absorbance, but in the text it is Fig2C is a reference to light microscopy and cell aggregates - so please correct that.
We would like to thank the reviewer for a careful and thorough reading of this manuscript and for the thoughtful comments and constructive suggestions, which help to improve the quality of our manuscript. Following the reviewer’s comments, we have corrected and improved the manuscript. We hope that you find our responses satisfactory and that the manuscript is now acceptable for publication.
Here is a point-by-point response to the reviewer's comments and concerns.
Reply: Thank you for this suggestion. We have corrected the mistake and provided new figure 2. Line 109, 112, Line 118-125.
Authors applied a lot of different techniques, which are usually suficciently described. I have few issues though:
Question: - in MIC determinations - did you check this "visible" effect by eye or in some reader? In 96-well plate testing MIC by eye would be tricky in my opinion - I would reccomend using plate reader to measure OD with blank control adjusted to starting cell suspention. Also have the bacterial suspentions were done in MH medium?
Reply: Thank you for your suggestion. We appreciate the reviewer’s comment. For the confirmation of MIC determination, we usually use 30 microlitre resazurin dye (0.015%) and incubated for 2 h (Elshikh et al., 2016). In this study, we did not show the data.
Reply: For preparing bacterial suspension, we used MH broth. (Line 298-304)
Question:- what concentrations of RK were used for growth curves?
Reply: For growth curve analysis, the concentration of RK ranges from 0 to 1000 μg/mL. (Line 307).
Question:- For protein extraction what medium was used to culture bacteria? LB?
Reply: For the protein extraction studies, we used LB without salt media. Line 353.
Question: Please, check the english for spelling and grammar: e.g., line 14 (abstract) not consist but contains (or consist of); line 293 (methods) bacterial solution was added.
Reply: Thank you for your suggestion. We corrected the mistakes. Line 14, Line 302
Question: The obtained results are fully discussed and the conclusions are supported by the results. It leaves an interesting scientific niche for further studies.
We also greatly appreciate the reviewers for their complimentary comments and suggestions.